# Filling the gaps between tide gauges: Demonstrating high-resolution seasonal high tide flooding predictions using NOAA's Coastal Ocean Reanalysis

Matthew P. Conlin[1,2]*, Gregory Dusek[1], John Ratcliff[1,2], John A. Callahan[1,2], Karen E. Kavanaugh[1], William Brooks[1], Blake Waring[1,3], Analise Keeney[1], William Sweet[1], Matthew J. Widlansky[4,5]

1 National Ocean Service, National Oceanic and Atmospheric Administration (NOAA), Silver Spring, Maryland, United States of America, 2 Ocean Associates, Inc., Arlington, Virginia, United States of America, 3 Consolidated Safety Services, Inc., Fairfax, Virginia, United States of America, 4 Cooperative Institute for Marine and Atmospheric Research, School of Ocean and Earth Science and Technology, University of Hawai'i at Mānoa, Honolulu, Hawai'i, United States of America, 5 Department of Oceanography, University of Hawai'i at Mānoa, Honolulu, Hawai'i, United States of America

* matthew.conlin@noaa.gov

## Abstract

High Tide Flooding (HTF) is a present and increasing hazard for coastal communities across the United States. NOAA provides HTF outlooks at U.S. tide gauges, however, many coastal communities lie relatively far from a tide gauge and therefore currently lack localized HTF guidance. In this study, we demonstrate an approach to generate spatially-continuous daily predictions of HTF at 400–500 m resolution out to a year into the future, by combining NOAA's monthly HTF outlook framework with the newly-released Coastal Ocean Reanalysis (CORA). Using CORA to derive daily HTF predictions at tide gauges, as compared to using gauge observations, results in average HTF model skill reduction of ≤5% using three different statistical metrics at one month lead time. Further, stations which obtain statistically skillful HTF predictions using gauge data also do so using CORA for 94% of cases. The results suggest that CORA could enable skillful HTF predictions away from tide gauges, supporting the possibility of providing high resolution HTF outlooks for much of the U.S. coastline. The potential value of these spatially continuous HTF predictions is illustrated by identifying communities near Charleston S.C. with different CORA-derived local HTF risk than that provided by the closest tide gauge. Finally, we describe outstanding questions and needs for the scaling of these results to an operational national-scale monthly HTF outlook.

purpose. The work is made available under the Creative Commons CC0 public domain dedication.

**Data availability statement:** All data used in this work are publically available from the National Oceanic and Atmospheric Administration. NWLON data can be accessed at: https://tidesandcurrents.noaa.gov/. CORA data can be accessed at: https://tidesandcurrents.noaa.gov/cora.html#resource.

**Funding:** Authors MC, JR, and JC are employed by Ocean Associates, Inc. Author BW is employed by Consolidated Safety Services, Inc. The funders provided support in the form of salaries for authors MC, JR, JC, and BW, but did not have any additional role in the study design, data collection and analysis, decision to publish, or preparation of the manuscript. The specific roles of these authors are articulated in the 'author contributions' section.

**Competing interests:** I have read the journal's policy and the authors of this manuscript have the following competing interests: Authors MC, JR, and JC are employed by Ocean Associates, Inc. Author BW is employed by Consolidated Safety Services, Inc. This does not alter our adherence to PLOS ONE policies on sharing data and materials.

## Introduction

High tide flooding (HTF), i.e., typically minor coastal flooding that can occur without a storm [1], is a present and increasing hazard for coastal communities across the United States. The height between impact-inducing flood levels and typical high tides continues to become smaller in many regions around the country experiencing relative sea level rise (SLR; [1]). Numerous studies have indicated that HTF already occurs regularly today [2–6] and will increase in frequency and severity with continued SLR in the coming decades [1,7–10]. Though immediate damage is not as impactful as major flooding due to coastal storms, the cost of recurrent HTF may be greater due to cumulative impacts to coastal infrastructure and economies. These impacts can include damage to transit infrastructure, reduced visits to impacted storefronts, damage to private property contributing to decreased real estate values, and degradation of wastewater treatment facilities [11–17].

In response to the threat of HTF, NOAA produces HTF outlooks at many tide gauges in the U.S. These span timeframes from annual outlooks supplemented by decadal projections that aid in budgeting and long-term planning [18]; to monthly outlooks that provide daily HTF probabilities from the present out to one-year to facilitate preparedness [19]. While the HTF outlooks are a critical advancement for preparedness and mitigation of HTF impacts, they are applied only to tide gauge observations from the National Water Level Observation Network (NWLON; [20]). NWLON sites are sparse and many coastal communities are far from a gauge. Further, due to small scale variability driven by local bathymetry/topography, riverine inputs, and other processes, water levels and flood thresholds at a NWLON site are not necessarily representative of locations even a short distance away [21,22].

To address these spatial gaps in data availability and flood guidance, NOAA initiated the Coastal Ocean Reanalysis (CORA), which provides 44 years of hourly water levels at 400–500 m resolution along the entirety of the U.S. Gulf and East Coasts (GEC) as well as the Caribbean [23,24]. The advent of CORA could make it possible to expand the HTF outlooks between and away from NWLON stations, which would enable unprecedented local HTF guidance across the U.S. These spatially continuous HTF outlook products would provide coastal communities far from NWLON stations with new, historical, and local information to aid in planning and flood mitigation efforts.

In this study, we demonstrate an approach to generate a spatially continuous monthly HTF outlook between tide gauges by leveraging CORA. We first compared CORA-derived HTF predictions with those from tide gauge observations at NWLON stations in the GEC to quantitatively assess the skill of CORA-derived HTF predictions at the stations. We then derived HTF predictions at shoreline-following sets of CORA nodes near Charleston, S.C. to demonstrate that spatially-continuous HTF predictions between NWLON stations can provide valuable localized HTF guidance.

## Background

### The seasonal HTF model and monthly HTF outlook

The seasonal HTF model (HTF model hereafter) was introduced by [25] and serves as the basis for NOAA's monthly HTF outlook, which is provided at many NWLON stations [19,26,27]. The HTF model combines tide predictions with climatologies of hourly non-tidal residuals (NTR; the difference between observed water levels and tide predictions), long-term linear trends of relative sea levels, and the damped persistence of monthly mean sea level (MSL) anomalies to provide daily predictions of HTF probability out to one year in the future. The NTR climatologies are developed by binning 23 years of hourly water levels by calendar month and tide decile and then adjusting for the linear long-term trend in MSL. The predicted NTR for any hourly timestep is given as a probability distribution that is assumed to be Gaussian with a mean of:

$$\mu_{clim}(t) = \mu_{month}(t) + \mu_{tide}(z) + p(t), \tag{1}$$

where $\mu_{month}(t)$ is the mean NTR for the calendar month in which time $t$ lies, $\mu_{tide}(z)$ is the mean NTR for the tide level decile in which predicted tide $z$ lies relative to the total mean NTR (termed the tide level adjustment factor in [25]), and $p(t)$ is the damped persistence value of the MSL anomaly to use for time $t$. Calculation of the standard deviation of the Gaussian $\sigma_{NTR}$ follows a similar convention. The predicted NTR distribution is then combined with the tide predictions and SLR trend and compared to an input flood threshold to determine the probability of flooding as the area under the model-predicted water level distribution that is above the flood threshold. Daily cumulative flood probabilities $P_{day}$ are then computed from the 24 corresponding hourly values $P(t)$ as the maximum hourly value that day $P_{max}$ plus the portion of each remaining value that day that is independent of the autocorrelation of the NTR signal $r(t)$:

$$P_{day} = P_{max} + \sum_{t=1}^{23} \left[ P(t) \left( 1 - r(t) \right) \right]. \tag{2}$$

[25] applied the HTF model to 98 NWLON gauges in the U.S. Without considering the persistence of the MSL signal ($p(t) = 0$, which they termed the "climatological model"), the HTF model was found to skillfully predict HTF days at 61 of the 92 gauges that experienced at least 10 HTF days using a Brier Skill Score (BSS) for a retrospective assessment. The performance of the HTF model was found to scale with tidal contribution to the total variance in the water level signal. This is because, while some information about temporal patterns of weather events can be retained in the NTR climatologies, flooding driven by individual weather events cannot be predicted by the model. Hence, model performance is weaker in locations like the Gulf, where the tidal signal is small compared to the contribution of weather events to flooding. Similarly, model performance was found to scale with the distance between the flood threshold and mean water level, as tidal contributions to flooding become increasingly important as the average daily high tide approaches the flood threshold. Inclusion of the MSL persistence (termed the "persistence model"), as used in this work, improved model performance primarily in the Pacific Islands and southern West coast.

### The Coastal Ocean Reanalysis (CORA)

The NWLON is the authoritative source for water level data in the U.S. and supports such crucial applications as maritime economic boundary delineation and safe and efficient marine navigation [20,28,29]. Large stretches of U.S. coastline, however, are relatively far from an NWLON station; in some cases hundreds of kilometers separate tide gauges. Therefore, many coastal communities do not have adequately representative NWLON water level observations. CORA was developed to fill these data gaps and provide more localized information about water levels, waves, and flooding [24].

CORA provides hourly water level data from 1979 through 2022 over an unstructured mesh that contains 1.8 million points for the Gulf and Northwestern Atlantic domain at typically 400–500 m resolution along the coast [23,24] (Fig 1). Detailed model setup and information is provided in [24] and will be summarized below. CORA is created using the two-dimensional barotropic ocean circulation model ADCIRC coupled with the spectral wave model SWAN. The open offshore boundary of the northwestern Atlantic model domain extends in an arc from Novia Scotia to Suriname, reaching eastward to the 55° W meridian. The model is forced with atmospheric pressure and 10 m wind velocities extracted from the ERA5 atmospheric reanalysis [30] as well as astronomical tides at the offshore boundary in the form of 10 principal tidal constituents extracted from TPXO [31]. Water level observations from 53 NWLON stations distributed throughout the GEC are low pass filtered at a 4-day cutoff period and dynamically assimilated into the model to capture non-barotropic sea level trends and variability. CORA therefore represents a reanalysis of local relative sea levels driven by both eustatic sea level changes and spatially variable rates of vertical land motion. Observations from 59 other NWLON stations in the GEC are used for model validation. Importantly, only open-coast stations are used in the data assimilation, meaning that validation (unassimilated) stations are in non-open coast (riverine, estuarine, and bay) environments.

[23] validated water levels from a preliminary version of CORA (version 0.9) compared to NWLON gauge observations. They assessed performance for GEC stations over the period 1979–2021, and found that long term linear trends, annual variability, monthly variability, and hourly nontidal residual variability from CORA water levels compared closely to observations. For example, the average linear trend from CORA between 1993 and 2020 was only 7% less than that from the gauges, while average monthly water level standard deviation from CORA was only 8% less than that from the gauges. Performance was generally stronger for the East Coast than the Gulf Coast, and weakest at stations far up rivers,

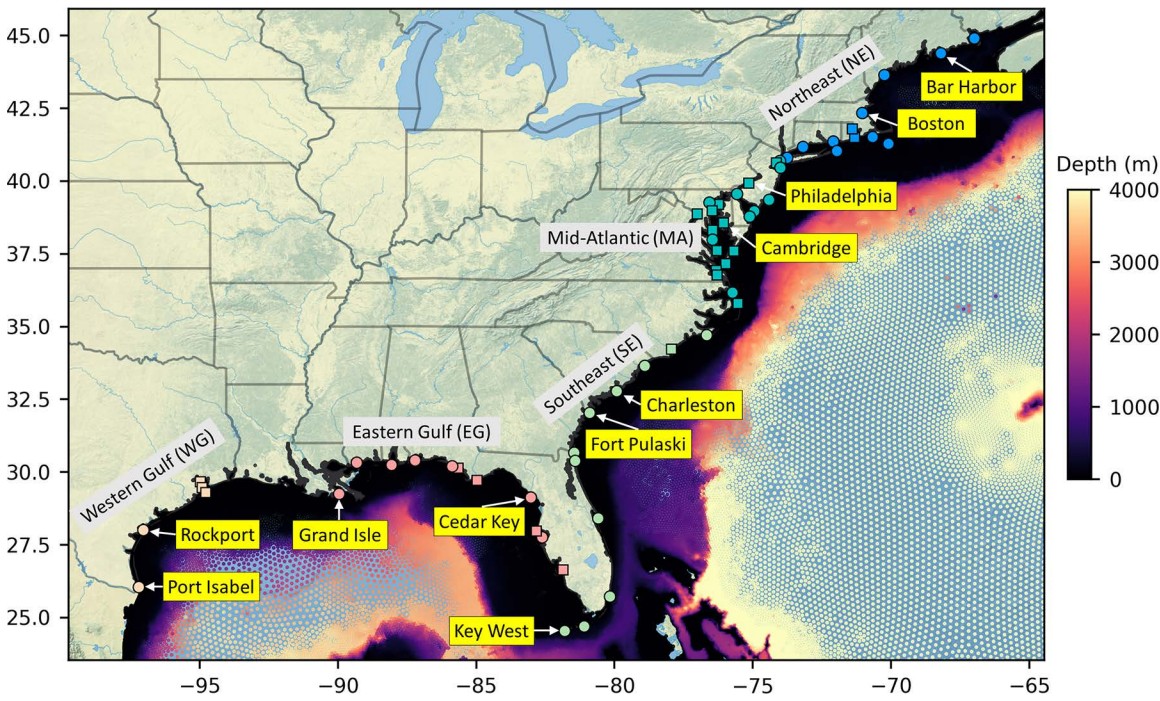

**Fig 1. Overview of the spatial domain of CORA and NWLON stations considered in this work.** Stations shown in later figures and/or specifically called out in the text are labelled. Stations depicted as circles (squares) are (are not) assimilated in CORA. The colors of the stations in each labelled region correspond to those used in later plots. Map source: Natural Earth.

as riverine processes are not included in the model. The transfer of spatially-varying CORA errors to CORA-derived HTF predictions is explored further in the Discussion section.

Here, we used a newer version of CORA (1.1) compared to what [23] assessed. The primary difference between the two versions (0.9 and 1.1) involves the data assimilation scheme, with improved performance in a few areas such as the Gulf Coast. [24] assessed the overall performance of CORA version 1.1 water levels relative to NWLON stations and found a RMSE of 0.15 m at validation (non-assimilated) stations and 0.11 m at assimilated stations.

## Methods

To understand the accuracy of CORA-derived HTF predictions, we first assessed differences between CORA-derived and gauge-derived HTF predictions at NWLON stations in the GEC. We then applied the HTF model to two subsets of shoreline-following CORA nodes around Charleston S.C. to demonstrate the potential value of high spatial resolution HTF predictions enabled by CORA.

### HTF model setup

Following [25], the HTF model was fit using 23 years of hourly water level observations and HTF predictions were completed at varying lead times. For example, the model was fit using hourly observations (from tide gauges and CORA nodes) from January 1, 1997 through December 31, 2019 (as in [25]), and then hourly HTF predictions were made for January 2020 (one month lead time), February 2020 (two month lead time), March 2020 (three month lead time), etc. The training period was then slid forward one month and the process repeated, such that February 2020 was predicted at 1 month lead time, etc. HTF predictions were made and evaluated for 2020–2022, representing the three most recent years currently available in CORA (note that [25] used a retrospective skill assessment over the training period). Three years were chosen as the evaluation period as a balance between capturing a relatively large number of flooding events and computational resources. In the comparison between gauge-derived and CORA-derived HTF predictions, we consider results at one month lead time only, though results at three month lead time, for example, were very similar (S1 Appendix).

### Preparation of CORA data

CORA water levels were obtained from the NOAA Open Data Dissemination platform [32]. Publicly available Python code notebooks were used to facilitate data access [33]. To compare with tide gauges, CORA water levels at the locations of the NWLON stations were derived using an inverse distance-weighted interpolation to the nodes comprising the mesh element encompassing the station, similar to [23]. We considered 61 stations in total. The CORA timeseries always indicated "wet" conditions (no missing data) at 58 of these 61 considered stations; at the remaining three stations- Apalachicola, F.L.; Bay Waveland Yacht Club, M.S.; and Port Isabel, T.X.- CORA data were dry < 0.1% of the training period. We considered the same NWLON stations as [25] except that we added two stations that were not used (Eastport, M.E. and Money Point, V.A.) and did not consider three stations with incomplete data records during the evaluation period (2020–2022; Naples, F.L., Sabine Pass, T.X., and Corpus Christi, T.X.). S2 Appendix provides details on these and all possible GEC NWLON stations. Of the 61 considered stations, more were assimilated in CORA (38) than were not (23). Assimilated stations are in open-coast environments, while non-assimilated stations are not.

For the Charleston, S.C. case study, CORA data from subsets of nodes following the coast were extracted and run through the HTF model as described in the HTF model setup section. The shoreline-following nodes were derived in different ways for the two spatial scales considered. Between the neighboring Fort Pulaski, G.A. and Charleston, S.C. NWLON stations, a vector layer was created of points 500 m from the shoreline- defined using NOAA's Continuously Updated Shoreline Product [34]- spaced at 1 km in the alongshore. After some manual refinement to ensure smoothness and that only CORA nodes that were nearly always inundated were included, CORA data were interpolated to these points using an inverse distance-weighted interpolation. Within Charleston Harbor and the immediate vicinity, a shoreline was

defined at the interface between nodes that always remain wet and those that do not. This interface was developed using an output file from each ADCIRC model year delineating nodes that were always inundated, and further refining to those that were on a mesh element which also contained a node that was not always inundated.

## Computation of physical quantities from CORA

In addition to water level observations, the HTF model relies on other physical quantities: datums, flood thresholds, SLR trends, and tide predictions. In general, standard methods are utilized through existing products to provide each of these quantities at NWLON stations [25,35,36]. However, these quantities must be computed at CORA nodes to provide input to the HTF model. Our approach is detailed below.

 **3.3.1. Datums and flood thresholds.** Many NWLON stations have established impact-based flood thresholds set by local National Weather Service Weather Forecast Offices, and multiple methods use these established flood thresholds to estimate impact-based flood thresholds anywhere [28,37,38]. Impact-based flood thresholds at NWLON stations in the GEC are typically between 0.5 and 0.6 m above mean higher high water (MHHW) using the method described in [28]. Since no single threshold describes all types of localized impacts, here we considered HTF predictions over multiple flood thresholds taken at 15 cm (~0.5 ft) increments from 0.15–0.60 m (0.5–2 ft) above MHHW. This approach aims to facilitate mapping of inundation impacts over a range of water levels and elicit a better sense of CORA-derived HTF prediction performance across multiple possible flood thresholds.

 While flood thresholds are relative to MHHW, CORA is referenced to the MSL datum. To relate CORA observations to MHHW, datums were computed at each CORA node from the CORA hourly water level timeseries using the First Reduction method in the Tidal Analysis Datums Calculator (TADC) [39]). The TADC identifies high and low waters from a low-pass filtered (at four cycles per day) water level signal and uses these to compute the standard tidal datums (MSL, MHHW, etc.). The computation was performed over the same 19-year National Tidal Datum Epoch used at NWLON stations (1983–2001). CORA water levels were placed onto the derived MHHW datum by setting this value as the zero-level of the timeseries.

 **3.3.2. Tide predictions.** Tide predictions serve as the core of the HTF model, as they provide a deterministic water level into the future that serves as the baseline atop which derived NTR climatologies and SLR trends are applied. At NWLON stations with data records longer than 19-years (all stations considered here), tide predictions are computed using harmonic analysis of hourly observations using at least 5 years for high-frequency harmonics and 19 years for low-frequency harmonics for a standard set of 37 harmonic constituents that contribute the majority of the tidal signal [35]. For CORA data, the Unified Tidal Analysis (UTide) software package [40] was used to perform the harmonic analysis and compute tidal constituents and tide predictions. Tidal constituents were computed from the hourly detrended CORA dataset using the 19 year period 2002–2020 with nodal corrections applied and with UTide able to determine the constituents to include based on a built-in signal to noise ratio analysis. Derived constituents were then used to reconstruct tide predictions over any period relative to MHHW.

 **3.3.3. Long-term relative sea level trends.** To adjust the (trendless) tide predictions for the observed SLR trend, [25] calculated a linear trend from 1980 through 2019 at each NWLON site following [36]. This method computes the trend using a lag 1 autoregressive linear model of monthly mean sea level (MSL) with the mean annual cycle removed. We used the same technique and time period to compute the long-term trend for each CORA node.

## Quantification of HTF model performance

Multiple metrics were used to quantitatively assess differences between CORA-derived and gauge-derived HTF predictions at NWLON stations. To assess the relative change in HTF predictions, we binned the HTF probabilities into hazard levels (HLs). The monthly HTF outlook discretizes HTF probabilities into three "likelihood" categories: unlikely for <5% probability, possible for 5–50% probability, and likely for ≥ 50% probability [19]. Here, we chose to use the HLs employed

by the NOAA Climate Prediction Center's Probabilistic Hazard Outlooks [41]: low risk of HTF for 0–20% probability, slight risk of HTF for 20–40% probability, moderate HTF risk for 40–60% probability, and high risk of HTF for 60–100% probability (all levels except the last are lower bound inclusive and upper bound exclusive). Using these HLs, we defined "HL agreement" as the percentage of daily CORA-derived HTF predictions that achieved the same HL as gauge-derived HTF predictions. Further, based on enumerating the HLs (e.g., 1 = low risk, 2 = slight risk, 3 = moderate risk, 4 = high risk), we also assessed the bias (mean error) and mean absolute error (MAE) of daily CORA-derived HTF predictions relative to gauge-derived HTF predictions. For example, a HL MAE of 0.1 indicates an average absolute HL error of 10% of a hazard level. These metrics provide practical guidance on the extent to which CORA-derived HTF predictions could yield the same decision-support information as gauge-derived HTF predictions.

Additionally, we utilized the continuous ranked probability score (CRPS) [42] to evaluate the performance of the model in terms of predicted water levels. The CRPS is effectively the squared area between the HTF model-derived water level Gaussian cumulative distribution function (CDF) and the step function representing the observed water level:

$$CRPS = \int_{-\infty}^{\infty} [F_f(\eta) - H(\eta - \eta_0)]^2 \delta\eta,$$

(3)

where $F_f$ is the model-derived CDF, $H$ is the Heaviside step function, and $\eta_0$ is the observed water level. The CRPS can be interpreted as the average model performance integrated across all possible HTF thresholds. We computed and compared the average CRPS of daily maximum water levels for the 2020–2022 prediction period for both gauge-derived and CORA-derived predictions.

Following [25], we also evaluated the performance of the HTF model- for both CORA and gauge input- in terms of predicted flood days. We identified the stations where the Brier Skill Score (BSS) is greater than the standard error of the BSS ($BSS_{SE}$), as computed following [43]. The BSS was computed as:

$$BSS = 1 - \frac{BS}{BS_{clim}},$$

(4)

where $BS$ is the Brier Score of the model predictions and $BS_{clim}$ is the $BS$ of HTF predictions made using the climatological mean observed probability at all timesteps [42]. $BS$ was computed as:

$$BS = \frac{1}{n} \sum_{t=1}^{n} (P(t) - o(t))^2,$$

(5)

where $n$ is the timeseries length, $t$ is the timestep, $P$ is the model-derived HTF probability, and $o$ is the observed flood value ($o(t) = 1$ if HTF occurred, $o(t) = 0$ if HTF did not occur). The stations for which $BSS > BSS_{SE}$, which are termed "skillful", were compared for CORA-derived and gauge-derived predictions. Additionally, to provide a more concise comparison, we computed the $BSS$ of CORA-derived HTF predictions using the gauge-derived HTF predictions as the reference model, which we term the relative Brier Skill Score (rBSS):

$$rBSS = 1 - \frac{BS_{CORA}}{BS_{gauge}}$$

(6)

The rBSS can be interpreted as a proportional performance change, according to the $BS$ metric, when using CORA as input vs. using gauge data as input to the HTF model. $rBSS < 0$ indicates weaker model performance using CORA ($BS_{CORA} > BS_{gauge}$), while $rBSS > 0$ indicates stronger model performance using CORA ($BS_{CORA} < BS_{gauge}$).

Finally, we also computed Relative/Receiver Operating Characteristic (ROC) curves for gauge-derived and CORA-derived HTF predictions [44,45]. ROC curves plot the true positive rate as a function of the false positive rate of HTF occurrence for all possible HTF probability warning levels (the model-derived HTF probability at or above which HTF is deemed to occur). For each flood threshold, an average curve was computed for each spatial region (see Fig 1) using both gauge and CORA input. We also computed the area under these curves (AUC) [44] as measures of model performance and quantified the decrease in AUC for CORA-derived HTF predictions relative to gauge-derived HTF predictions. $AUC = 1$ indicates perfect model performance, while a random guess obtains $AUC = 0.5$.

## Results

### CORA datums, tide predictions, and SLR trends

CORA and gauge MHHW datums were highly correlated ($r = 0.99$) with a root mean squared error (RMSE) of 8 cm and slight overall negative bias of 4 cm (Fig 2a). For the period 1997–2022, the RMSE of hourly CORA-derived tide predictions relative to the published values at the NWLON stations was 18 cm, though was 11 cm for tides above MHHW (those responsible for HTF; Fig 2b). CORA trends were relatively less correlated with the gauges ($r = 0.65$). The RMSE of CORA-derived trends relative to the published values at the NWLON stations was 1.27 mm/yr (Fig 2c), however not considering Eagle Point, TX the RMSE was 0.74 mm/yr. Eagle Point, TX has been strongly impacted by land subsidence due to oil and gas extraction [46], processes CORA does not capture at this unassimilated station (Fig 2c).

### Accuracy of CORA-derived HTF predictions

In general, CORA produced similar HTF predictions to the gauges at the NWLON stations, and in particular captured the same peak events. For example, Fig 3 shows example timeseries of daily HTF probability above $MHHW + 0.30$ m over the prediction period at one-month lead for selected stations, where the agreement between gauge-derived and CORA-derived HTF predictions is qualitatively clear. There were, however, cases of underprediction, particularly for some of the highest-probability peaks derived from the gauge at Port Isabel, TX (Fig 3f). Cases of overprediction were also apparent, and can be seen at all example stations in Fig 3.

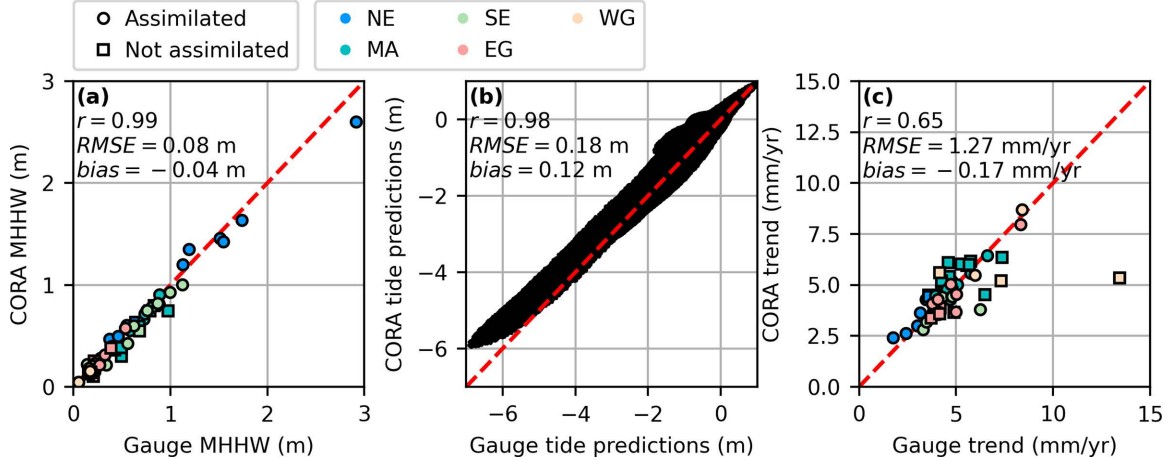

**Fig 2. Comparison of CORA-derived physical quantities with those published at the NWLON stations used in this work. (a)** MHHW elevation relative to MSL, **(b)** Tide predictions relative to MHHW, and **(c)** linear relative sea level trend.

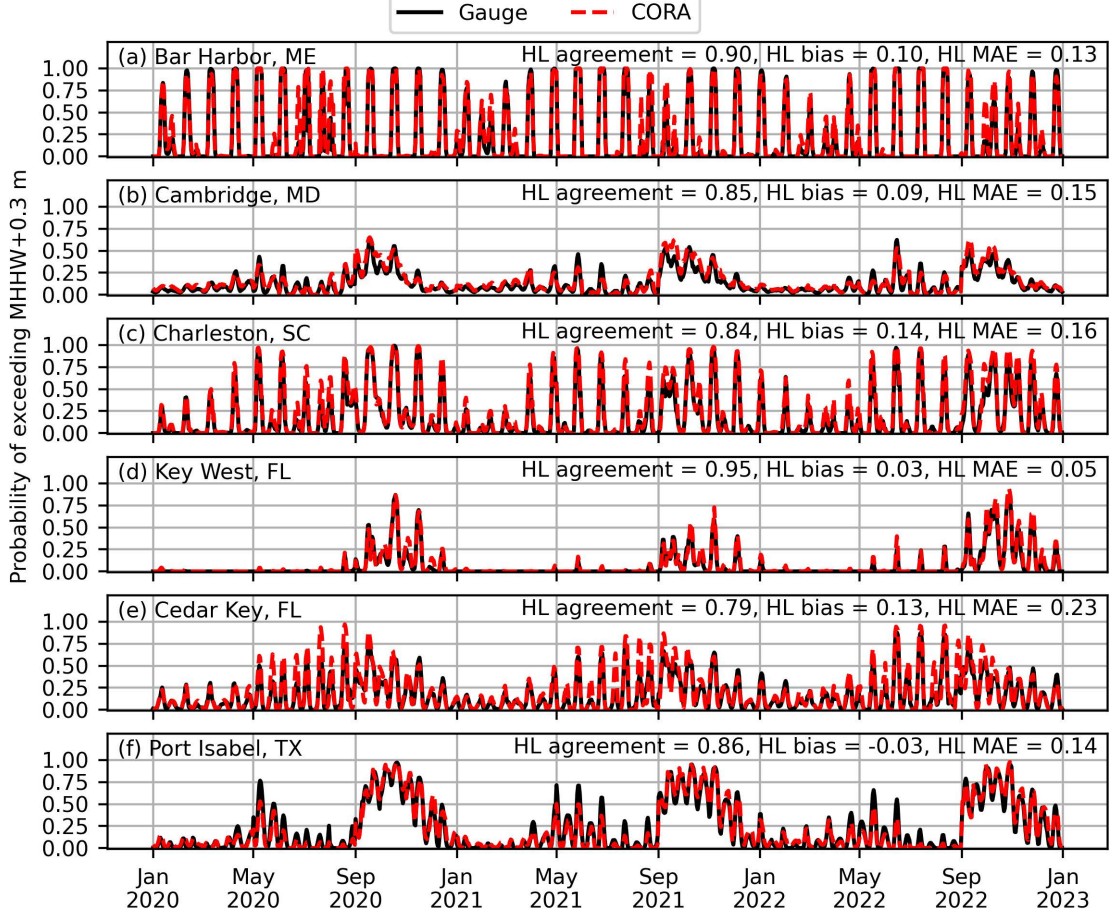

**Fig 3. Selected examples of gauge-derived (black) and CORA-derived (red) daily high tide flooding probabilities at one-month lead above MHHW + 0.30 m.** At least one station from each of the five geographic regions (see Fig 1) is shown. Hazard level statistics for each station are also shown.

Quantitatively, CORA produced the same HL as the gauges the majority of the time, as shown in Fig 4 and Table 1 and for all stations individually in S3 Appendix. For flood thresholds ≥ *MHHW* + 0.45 m, HL agreement was ≥ 84% on average within each region (Fig 4a, Table 1), while average absolute HL biases were ≤ 0.09 (Fig 4b, Table 1). HL agreement (Fig 4a), bias (Fig 4b), and MAE (Fig 4c) were inversely related to flood threshold: HL agreement increased from an overall average value of 77% for a flood threshold of *MHHW* + 0.15 m to 99% for a flood threshold of *MHHW* + 0.60 m, while bias decreased from 0.08 to 0.00 and MAE decreased from 0.26 to 0.01 (Table 1). HL biases tended to be slightly positive (CORA yielding increased HL relative to the gauge) on average across flood thresholds by <0.08, except the WG where the average bias across flood thresholds was negative at −0.09 (Table 1). Note that all values given above and below exclude Grand Isle, LA and Rockport, TX, where errors in CORA water levels and tide predictions led to strong overprediction of HTF probabilities (S4 Appendix).

Using the CRPS, performance of the HTF model declined by just 5% on average when CORA input was used instead of gauge data (Fig 5). On average, the largest performance degradation was found in the EG (8%) and the smallest was found in the MA (2%; 5–7% in the NE, SE, and WG). At 11 of the 61 stations, primarily in the MA (5 out of 11), CORA input yielded stronger HTF model performance than gauge data. The average increase in performance at these stations was only 3%.

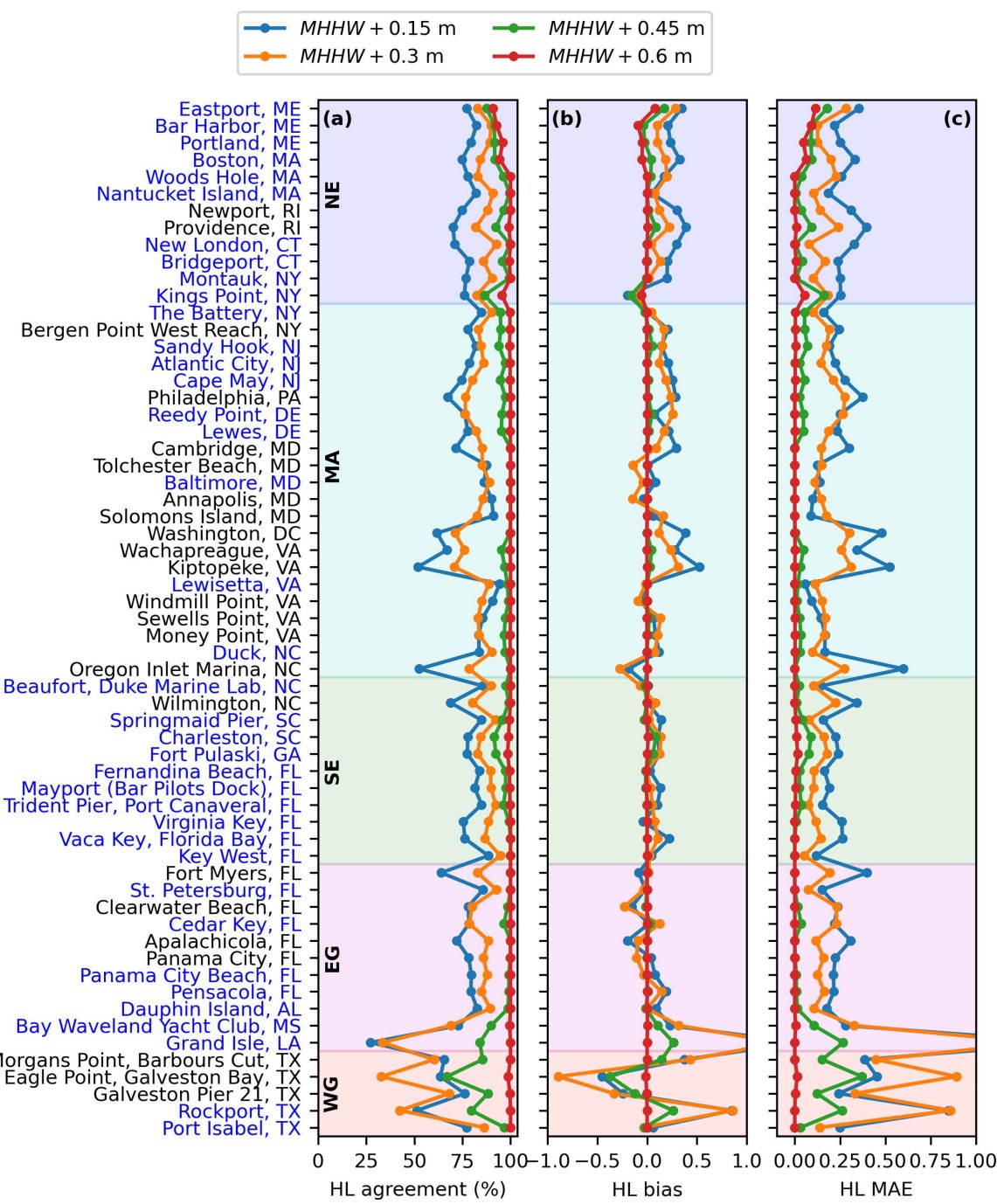

**Fig 4. Comparison of high tide flooding hazard levels (HL) derived from CORA relative to those derived from gauges for all flood thresholds considered over the period 2020 through 2022 at one month lead. (a)** The percentage of daily CORA-derived high tide flooding predictions that obtain the same HL as those derived from the gauge. **(b-c)** The bias **(b)** and mean absolute error **(c)** in enumerated daily HL for CORA-derived high tide flooding predictions relative to those from the gauge. The geographic regions (see Fig 1) are shown by the shading and labeled. Station names written in blue (black) are (are not) assimilated in CORA.

**Table 1. Average values of HL agreement, bias, and MAE, respectively, within each region (see Fig 1) and for each considered flood threshold at one month lead.**

| Region | N[1] | *MHHW* + 0.15 m | *MHHW* + 0.30 m | *MHHW* + 0.45 m | *MHHW* + 0.60 m | mean |
|---|---|---|---|---|---|---|
| NE | 12 (10,2)[1] | 77% \| 0.21 \| 0.28 | 87% \| 0.11 \| 0.16 | 94% \| 0.01 \| 0.07 | 97% \| −0.01 \| 0.03 | **89% \| 0.08 \| 0.14** |
| MA | 22 (9,13)[1] | 78% \| 0.14 \| 0.24 | 82% \| 0.09 \| 0.19 | 97% \| 0.01 \| 0.03 | 100% \| 0.00 \| 0.00 | **89% \| 0.06 \| 0.11** |
| SE | 11 (10,1)[1] | 80% \| 0.08 \| 0.21 | 88% \| 0.05 \| 0.12 | 97% \| 0.01 \| 0.03 | 100% \| 0.00 \| 0.00 | **91% \| 0.04 \| 0.09** |
| EG | 10 (6,4)[1] | 77% \| 0.02 \| 0.24 | 84% \| 0.01 \| 0.17 | 98% \| 0.01 \| 0.02 | 100% \| 0.00 \| 0.00 | **90% \| 0.01 \| 0.11** |
| WG | 4 (1,3)[1] | 71% \| −0.07 \| 0.33 | 62% \| −0.21 \| 0.45 | 84% \| −0.09 \| 0.17 | 100% \| 0.00 \| 0.00 | **79% \| −0.09 \| 0.24** |
| **mean** | | **77% \| 0.08 \| 0.26** | **81% \| 0.01 \| 0.22** | **94% \| −0.01 \| 0.06** | **99% \| 0.00 \| 0.01** | |

[1]N represents the number of NWLON stations used for the comparison, with values in parenthesis representing the number of stations that are and are not assimilated in CORA, respectively.

Additionally, CORA-derived and gauge-derived HTF predictions were skillful at nearly all the same stations using the metric of Dusek et al. (2022; Fig 6). Across the considered flood thresholds, CORA-derived HTF predictions were skillful at 94% of the stations for which gauge-derived predictions were also skillful, while there were an additional six instances where CORA-derived HTF predictions were skillful but those from the gauge were not. For the stations where gauge-derived pre-dictions were skillful, those from CORA were also skillful at 58/58, 54/57, 35/41, and 20/21 stations (20/20 for stations with at least 10 floods) for the considered flood thresholds in increasing order (Fig 6a-d). This result also indicates that HTF model performance decreased with increasing flood threshold; fewer stations were skillful for both gauge and CORA input as flood threshold increased. The greatest loss of skillfulness for CORA input, in terms of *BSS*, was in the MA and Gulf regions (Fig 6a-d).

Summarizing these differences in *BSS* using the *rBSS* (Fig 6e), performance of the HTF model declined by just 2% (*rBSS* = –0.02) on average across regions and thresholds when CORA was used instead of gauge data. |*rBSS*| was inversely proportional to flood threshold, with average values of −0.05 for *MHHW* + 0.15 m, −0.03 for *MHHW* + 0.30 m, −0.01 for *MHHW* + 0.45 m, and 0.00 for *MHHW* + 0.60 m. Across flood thresholds, 91% of the 236 total datapoints obtained |*rBSS*| ≤ 0.10. 39% obtained *rBSS* > 0, i.e., an increase in HTF model performance for CORA input relative to gauge input.

The ROC curves also confirmed that CORA-derived HTF predictions were quite similar to those from the gauges, with the curves from CORA and gauge typically lying nearly atop one another (Fig 7a-d). The average reduction in AUC for CORA input was only 1% as compared to gauge input (Fig 7e). AUC reductions for all regions and thresholds were ≤ 4% except in the EG for *MHHW* + 0.30 m (7%). Similar to the CRPS (Fig 5) and BSS (Fig 6) analyses, there were instances for which HTF model performance was stronger (larger AUC) for CORA input than gauge input, particularly for a HTF threshold of *MHHW* + 0.45 m (Fig 7a-d). Similar to the BSS analysis (Fig 6a-d), HTF model performance as a whole decreased as the flood threshold increased, with ROC curves becoming closer to the 1:1 random guess line (Fig 7a-d). Further, for all HTF thresholds model performance for both CORA and gauge input was highest in the NE, lower in the MA and SE, and weakest in the Gulf regions.

## Case study near Charleston, S.C

A demonstration of the value of CORA-derived HTF predictions was performed near Charleston, S.C. for two spatial scales at a flood threshold of *MHHW* + 0.60 m (Fig 8). HTF predictions were made around two adjacent barrier islands (Fig 8b) and between the neighboring NWLON stations of Charleston and Fort Pulaski (Fig 8c). Note that, for this flood threshold, HL agreement is > 98% and HL bias is ≤ 0.01 at these stations (Fig 4a,b).

Both applications illustrate spatial variability in HTF risk that is not observable using tide gauges alone. For example, certain bay- and ocean-facing sides of the barrier islands were predicted to obtain different HLs on this day (Fig 8b), as

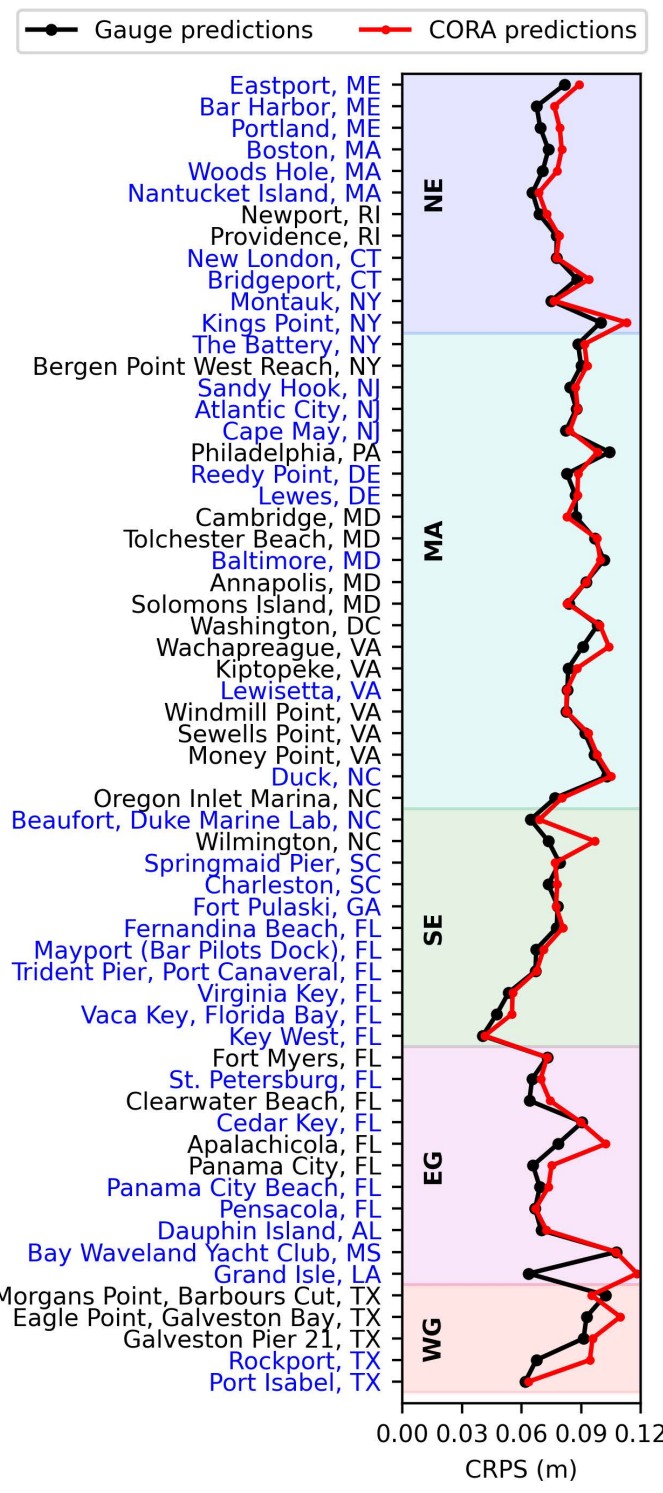

**Fig 5. Continuous Ranked Probability Score (CRPS) of daily maximum water levels over the period 2020 through 2022 at one month lead.**
Gauge-derived (CORA-derived) probabilistic water level predictions are shown in black (red). The geographic regions (see Fig 1) are shown by the shading and labeled. Station names written in blue (black) are (are not) assimilated in CORA.

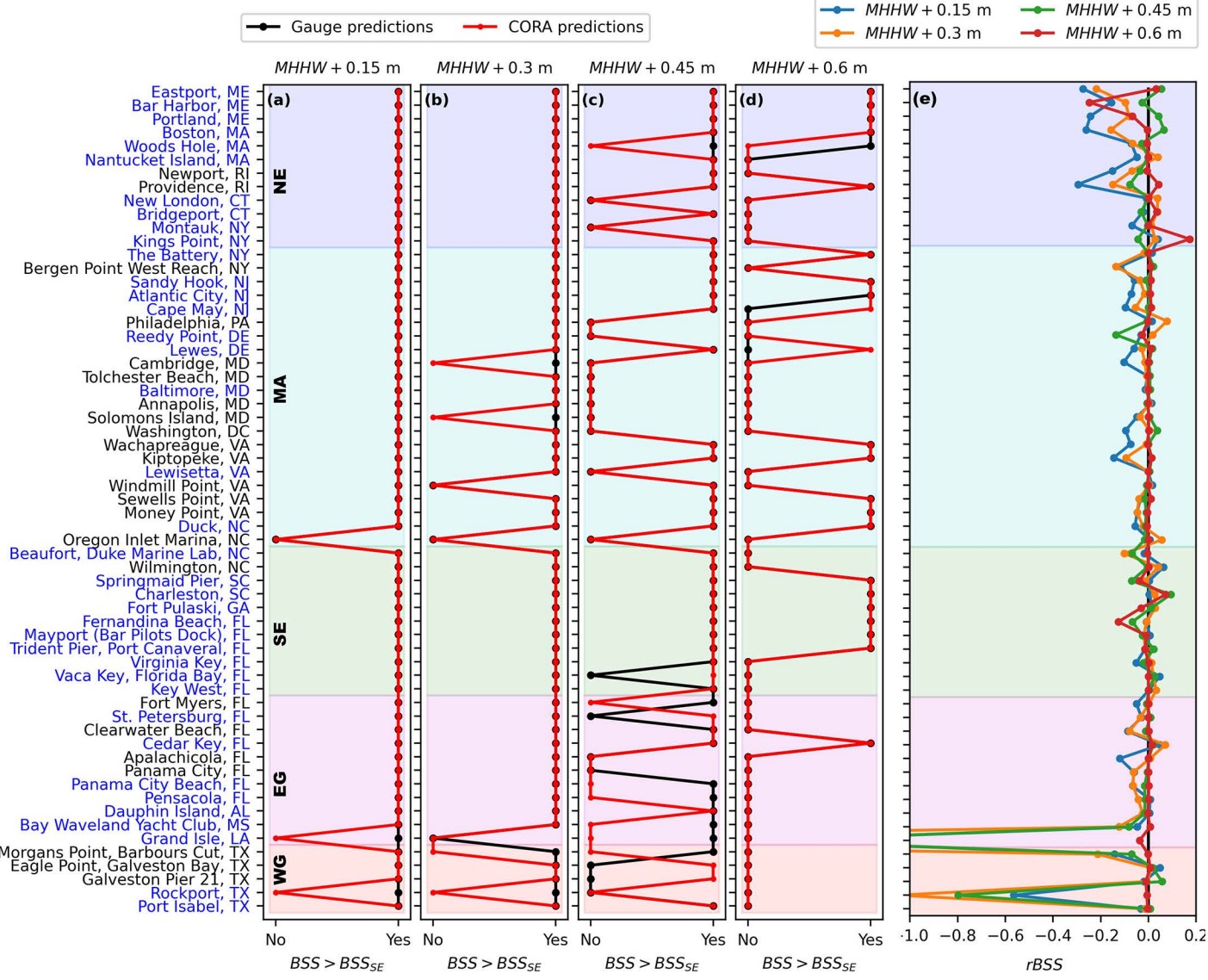

**Fig 6. Brier Skill Score comparisons for CORA- and gauge-derived HTF predictions. (a-d)** Comparison, at one-month lead, of stations that obtain a Brier Skill Score (BSS) that is greater than the standard error of the BSS as computed following Bradley et al. (2008) for both gauge-derived (black) and CORA-derived (red) high tide flooding predictions for flood thresholds of **(a)** *MHHW* + 0.15 m, **(b)** *MHHW* + 0.30 m, **(c)** *MHHW* + 0.45 m, and **(d)** *MHHW* + 0.60 m. **(e)** *rBSS*, interpretable as the proportional performance change in terms of Brier Score when using CORA as input relative to using gauge data as input, for all stations and flood thresholds considered at one-month lead. The geographic regions (see Fig 1) are shown by the shading and labeled. Station names written in blue (black) are (are not) assimilated in CORA.

could be expected based on hydrodynamic differences around a barrier island driven by patterns in bathymetry, wave dissipation, constrictions to exchange, and shoreline orientation [47–49]. Similarly, all ocean-facing locations between the Charleston and Fort Pulaski tide gauges were predicted to obtain the same HL as the Fort Pulaski gauge, even those that lie closer to the Charleston gauge (Fig 8c). The difference at ocean-facing locations near Charleston may result from the gauge's inland position within the harbor, which modifies tidal and water level dynamics relative to the open coast. This localized HTF guidance could help these communities more effectively focus resources when preparing for potential

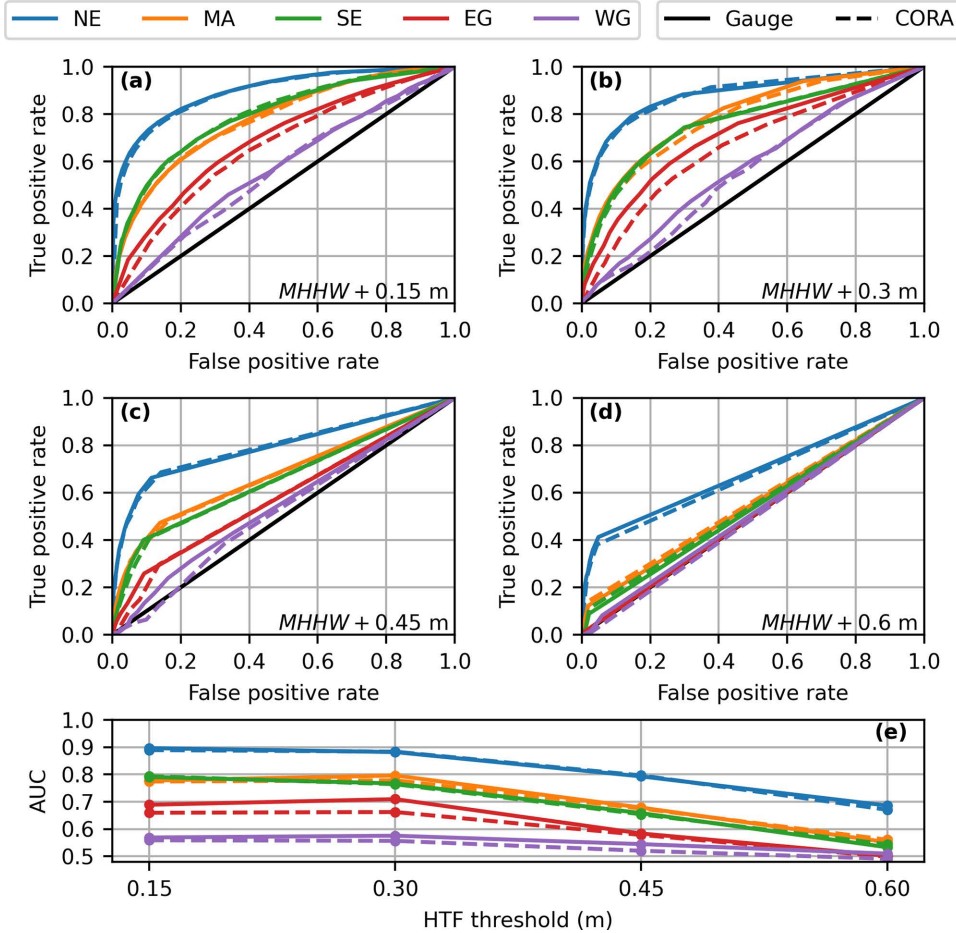

**Fig 7. Receiver Operating Characteristic (ROC) curves and area under the curves (AUC) for high tide flooding predictions at one month lead.**
**(a-d)** ROC curves for flood thresholds of **(a)** *MHHW* + 0.15 m, **(b)** *MHHW* + 0.30 m, **(c)** *MHHW* + 0.45 m, and **(d)** *MHHW* + 0.60 m. **(e)** AUC for each curve. Note that AUC = 1 indicates perfect predictions, while a random guess obtains AUC = 0.5.

flooding events, rather than having to rely on the nearest tide gauge which may be 10s of km away and/or in an area with considerably different morphodynamics.

## Discussion

Our results indicate that there was minimal change in HTF predictions at NWLON stations when CORA was used in place of gauge observations. Average reduction in HTF model performance was just 5% using the CRPS (Fig 5), 2% using the *rBSS* (Fig 6e), and 1% using the AUC (Fig 7). Additionally, CORA yielded the same HL as the gauge at least 77% of the time on average for each flood threshold, with greater agreement at the highest considered flood thresholds (Fig 4, Table 1). This minimal performance change translated to similar, though not identical, stations that were skillful when using CORA as compared to gauge observations: stations that were skillful using gauge observations were also skillful using CORA for 94% of cases across the considered flood thresholds (Fig 6a-d). Additionally, it is noteworthy that CORA and gauge input both yielded the same pattern of decreasing HTF model performance with increasing flood threshold (Figs 5,6), as also documented in [25]. This pattern is because, at higher flood thresholds, the tidal contribution to floods is relatively small, and there are fewer floods overall on which to assess model skill in terms of flood days.

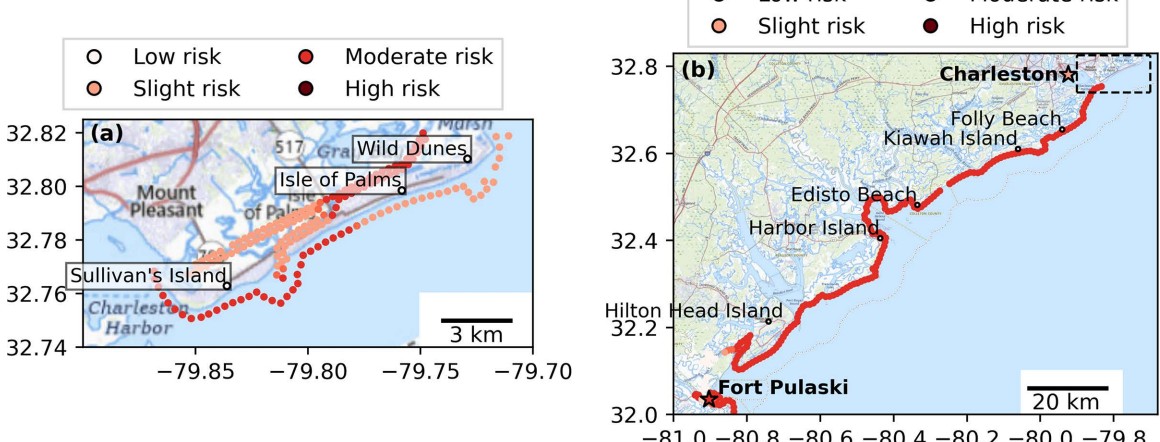

**Fig 8. Demonstration of spatially continuous predictions of risk of high tide flooding above *MHHW*+ 0.60 m (*MHHW* + 2 ft) at one-month lead time using CORA on October 19, 2020. (a)** HTF hazard level predictions around two barrier islands near Charleston using a 5-point nearest neighbor smoothing. **(b)** HTF hazard level predictions for the coastline between the Fort Pulaski and Charleston tide gauges using a 5-point nearest neighbor smoothing, with predictions from the tide gauges shown as stars. The dashed box in (b) shows the spatial area of **(a)**. The hazard levels have the following HTF probability bounds: low: 0-20%, slight: 20-40%, moderate: 40-60%, high: 60-100%. Map source: USGS National Map.

The minimal HTF model skill reduction of ≤5% on average for CORA-based HTF predictions using three different statistical metrics yields some confidence that CORA can provide spatially-continuous HTF guidance away from the stations. Considering the accuracy metrics at only the stations not assimilated in CORA (those with names in black in Figs 3–6) provides a proxy for performance away from the gauges. In particular, since unassimilated stations are located in non-open coast environments [24] (see also Fig 1), accuracy at unassimilated stations may be somewhat representative of accuracy specifically in these environments. The comparison between CORA-derived and gauge-derived HTF predictions at unassimilated stations was similar to assimilated stations. For example, the change in CRPS at the unassimilated stations was 6%, similar to that at assimilated stations of 4% (5% overall; Fig 5). Additionally, HL agreement at unassimilated stations was, on average, within 10% of that at assimilated stations for all flood thresholds considered, at 80%, 87%, 96%, and 99% for the flood thresholds in increasing order at assimilated stations, and 73%, 78%, 96%, and 100% at unassimilated stations (Fig 4). Finally, of the eight stations which were skillful using gauge input but not for CORA input for at least one flood threshold, four were assimilated and four were not (Fig 6).

While the comparison between CORA-derived and gauge-derived HTF predictions at the NWLON stations indicates minimal performance change overall, there is variability as a function of region and HTF threshold. For example, the Gulf regions showed the greatest decrease in number of stations that were mutually skillful as the flood threshold increased (Fig 6). The EG also obtained the maximum single-threshold AUC reduction for any region or threshold at 7%, as well as the largest region-averaged reduction in CRPS at 8%. It is noteworthy that there are known limitations to CORA in the Gulf: available data from NWLON stations to drive the data assimilation are particularly scarce here (see Fig. 23 in [24]), causing water level errors relative to typical variability to be especially large (see Fig. 27 in [24]). Indeed, the WG has only four comparison stations, of which only one is assimilated in CORA (Table 1; S3 Appendix; see also Figs 4-6). Additionally, there are Gulf locations with relatively large local rates of land subsidence (such as Eagle Point, T.X.; [46]), which CORA cannot capture at unassimilated locations. Grand Isle and Rockport in particular (S4 Appendix), as well as the difference in trend at Eagle Point (Fig 2c), highlight these inaccuracies in CORA.

The variations in the comparison between CORA- and gauge-derived HTF predictions as a function of region and threshold are also partially due to interacting biases between the HTF model and CORA over different flood thresholds. At Philadelphia, for example (Fig 9a), CORA-derived HTF probabilities tend to be slightly higher than those from the gauge (HL biases of 0.29, 0.24, 0.01, 0.00 for the four flood thresholds in increasing order; Fig 4b). This high bias for CORA somewhat counteracted a structural underprediction of the HTF model- due to its inability to capture weather driven flooding [25]- and caused the station to have a smaller CRPS for CORA input (Fig 5) and a positive or zero *rBSS* for all flood thresholds (Fig 6e). This helps explain how some stations achieved stronger model performance for CORA input. For the example of Boston (Fig 9b), however, where CORA-derived HTF probabilities tended to be similarly high-biased, HTF probabilities were so large during peak events that the high-bias only contributed to increasing errors during periods of high predicted probability without an observed flood (Fig 9b). This helps explain the relatively large performance degradation in terms of *rBSS* and CRPS in the NE, particularly for lower flood thresholds (Figs 4,5e). Slight differences in methods for the computation of physical quantities for gauges vs. CORA (e.g., datums, tide predictions, and long-term trends; Fig 2) could also explain some differences in HTF predictions.

## Towards an operational community-level HTF outlook

By documenting minimal change in HTF predictions and skill using CORA at the NWLON stations and demonstrating the potential output and value of localized HTF predictions near Charleston, S.C., this work lays the foundation for an operational spatially continuous monthly HTF outlook that can provide localized flood likelihoods nationwide. A number of outstanding questions remain in order to scale these results to a national operational product. Most importantly, the current methodology and data do not support real-time predictions, as the HTF model requires water level observations up to and including the month before HTF predictions are to be made so that damped persistence values can be computed and prediction lead times are small [25] (see also the Background section). More frequent, monthly to yearly updates of CORA, similar to other ocean and atmosphere reanalyses (e.g., ERA5; [30]) would be perhaps the simplest approach to allow real-time predictions. Alternative approaches could also be pursued with the existing CORA data, such as developing

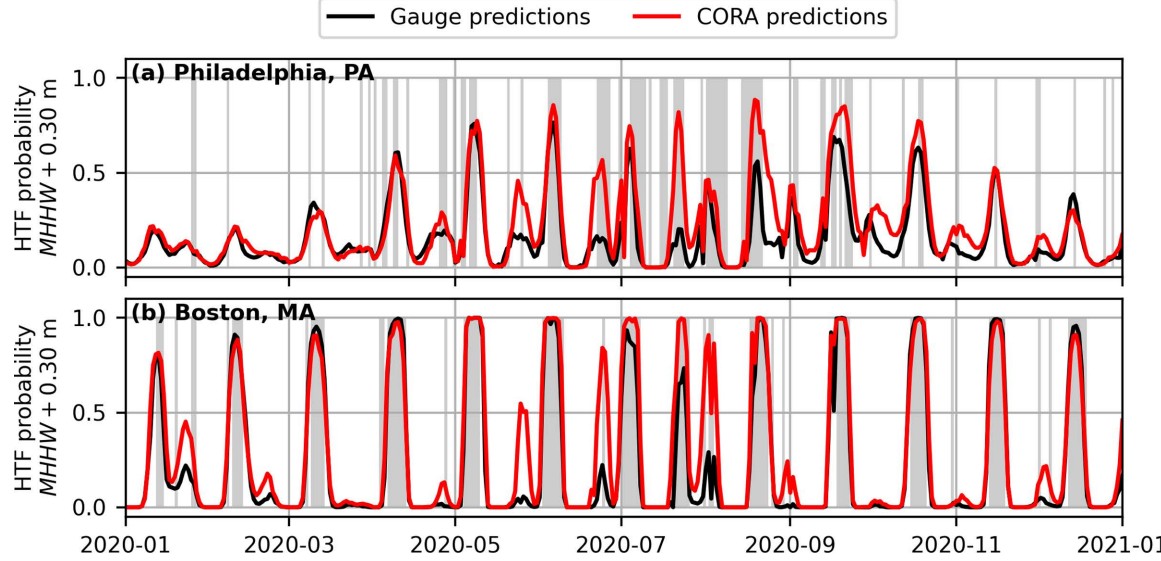

**Fig 9. Comparison of HTF predictions and observations above *MHHW + 0.30 m* for two example stations. (a)** Philadelphia, P.A. and **(b)** Boston, M.A. Observed floods are indicated by gray shading. Only observations and HTF predictions for 2020 are shown for visualization purposes; predictions were made for 2020 through 2022.

relationships between CORA nodes and tide gauges for the time period covered by CORA, either through statistical or machine-learning approaches, and extending these to real-time predictions. Further research could focus on this topic. Relatedly, efforts are ongoing to replace the sea level persistence in the HTF model with downscaled climate model output [50], which could provide further flexibility for real-time predictions.

Further, the comparison at NWLON stations, even at those that are not assimilated in CORA, does not provide a fully comprehensive understanding of CORA-derived HTF prediction accuracy in all relevant coastal locations and morphodynamic environments. For example, even though non-assimilated stations are in non-open coast environments, they do not sample back-barrier locations or locations very far up rivers, leaving the accuracy of CORA-derived HTF predictions in these environments technically unknown. While potential validation data in these and other environments is inherently limited by a lack of observations (hence the need for CORA), non-NWLON tide gauge networks such as those recently available through Hohonu Inc. [51] as well as supporting data such as media reports of flooding occurrences could be used to more fully assess CORA-derived HTF prediction accuracy at new locations in future work. Additionally, a more bottom-up approach could be an analysis of pre- and post-assimilated CORA output to deduce a more robust understanding of the spatially variable effects of the data assimilation as a function of morphodynamic environment and/or distance from NWLON stations. The 400–500 m resolution of CORA is also an important consideration: it is possible that narrow waterways such as back-barrier bays and inland estuaries with hydrodynamics varying on spatial scales of O(1–10 m) may not be captured sufficiently or at all by CORA, leaving HTF predictions not possible in these locations. Improving the spatial resolution in future CORA versions could help alleviate this limitation. Even if sampled, however, such environments may not be inundated at all times and are likely very shallow, which could drive nonlinear interactions between water level components that result in strongly non-Gaussian distributions [52]. Since the HTF model was developed for always-inundated tide gauge data, we here limited the application to CORA nodes that were (nearly) never dry. Further research is needed to develop methods to handle intermittently dry data and further explore possible non-Gaussian distributions in these shallow CORA locations. Indeed, it has recently been shown that non-Gaussian stochastically generated skewed distributions may better characterize non-tidal residuals than Gaussian distributions at NWLON stations [53].

A further challenge for a national-scale product will be the delineation of a reliable and useful shoreline-following subset of CORA points at which to deliver HTF predictions at a national scale.. The two techniques applied at prototype-scale in this work- tracing points delineating the boundary of always/not always wet nodes (Fig 9b) and utilizing a buffer from a shoreline model (Fig 9c)- are promising. However, these approaches will likely require manual refinement or additional local considerations at a larger scale; for example around the highly complex coastlines of Maine and the Mississippi River delta. Other approaches could also be viable, such as tracing a depth contour for nodes that are inundated a certain percentage of the time, and further research is needed on this topic.

Finally, it will be useful to provide uncertainty bounds on HTF predictions, as this helps convey input and model uncertainties to potentially non-expert end users. We have explored a method to propagate uncertainty in CORA water levels, which are on the order of 0.10–0.15 m [24], to CORA-derived HTF predictions. The method is based on a Monte Carlo approach, wherein many realizations of model predictions are generated when randomly sampling model parameters from a probability distribution capturing their uncertainties [54]. Using observed error distributions in CORA water levels and tide predictions, an ensemble of possible HTF predictions can be made by varying CORA values within these distributions (e.g., Fig 10 red area). For example, by examining CORA vs. observed NTR, Eq. 1 can be modified as:

$$\mu_{clim}(t) = \mu_{month}(t) + \mu_{tide}(z) + p(t) \pm \delta_{NTR}, \tag{7}$$

where $\delta_{NTR}$ is a random value between 0 and the standard deviation of the CORA NTR errors. The modified value of $\mu_{clim}(t)$ is then compared with the distance between the flood threshold and predicted tide similarly modified within the standard deviation of the CORA tide prediction errors $\delta_{tide}$. Comparing to Fig 3c, it is clear that while deterministic

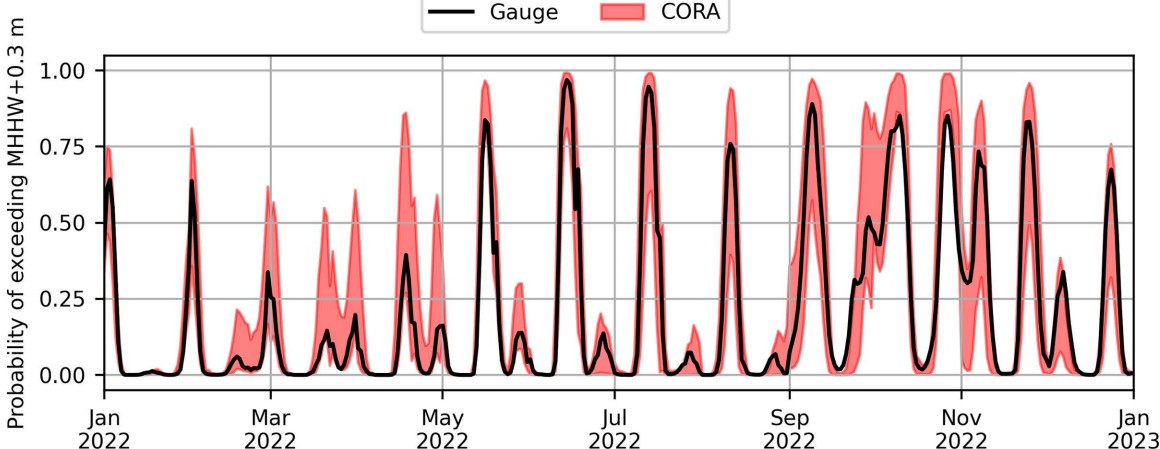

**Fig 10. Demonstration of probabilistic CORA-derived high tide flooding predictions.** CORA-derived high tide flooding probability uncertainty bounds above *MHHW* + 0.30 m for the Charleston NWLON station at one month lead are shown in red, in comparison to gauge-derived predictions show in black. Uncertainty bounds show the 25th-75th percentile of a 20-member ensemble. Only predictions for 2022 are shown for visualization purposes; predictions were made for 2020 through 2022.

CORA-derived HTF predictions are often higher than those derived from the gauge for this example, the (temporally-varying) uncertainty range on CORA-derived HTF predictions always encompasses the gauge predictions (Fig 10). For example, for the high probability event at the beginning of July 2022, gauge-derived HTF probability is 0.94, while the bounds of CORA-derived HTF probability are 0.60–0.99 (Fig 10). This approach could provide a useful method for delivering probabilistic HTF predictions with uncertainty estimates. However, the iterative nature of the method is relatively inefficient, and other approaches should also be investigated to facilitate application at the many thousands of possible CORA nodes.

## Conclusions

The existing monthly HTF outlook, delivered only at NWLON stations, is an important tool to help local planners time the allocation of response staff and resources to areas with the highest flood risk. This work lays the foundation for a spatially-continuous monthly HTF outlook every 400–500 m using CORA by demonstrating minimal change in HTF predictions when CORA is used in place of gauge observations at NWLON stations, including those that are not assimilated in CORA, and illustrating cases of variable CORA-derived flood likelihoods along the coast not observable using gauges. While further research is needed to more comprehensively understand the accuracy of CORA-derived predictions in all relevant locations and morphodynamic environments, and challenges remain for scaling these results to a national operational product, a spatially continuous monthly HTF outlook will provide critical localized information to communities and empower more effective flood preparation and mitigation.

## Supporting information

**S1 Appendix. CORA-derived vs. gauge-derived HTF predictions at three-month lead.**
(DOCX)

**S2 Appendix. Overview of all GEC NWLON stations and those used in this study.**
(XLSX)

**S3 Appendix. CORA-derived HTF prediction performance results, at one month lead, for each station included in the analysis.**
(XLSX)

**S4. Appendix. Explanation of CORA errors at Grand Isle, L.A. and Rockport, T.X.**
(DOCX)

## Acknowledgments

The authors wish to thank the many individuals who contributed to the development of CORA, particularly those at the University of North Carolina's Renaissance Computing Institute and Tetra Tech/RPS. The authors also wish to thank Oscar Guzman for creating and sharing the vector layer-based CORA shoreline data. Finally, the authors wish to thank two anonymous reviewers whose comments improved the quality of the manuscript.

## Author contributions

**Conceptualization:** Gregory Dusek.

**Data curation:** John Ratcliff, William Brooks, Analise Keeney.

**Formal analysis:** Matthew Conlin, John Ratcliff.

**Investigation:** Matthew Conlin, Gregory Dusek, John Ratcliff, John A. Callahan, Karen E. Kavanaugh, Analise Keeney, William Sweet, Matthew J. Widlansky.

**Methodology:** Matthew Conlin, Gregory Dusek, John Ratcliff, John A. Callahan, Karen E. Kavanaugh, Blake Waring.

**Software:** Matthew Conlin.

**Supervision:** Gregory Dusek.

**Visualization:** Matthew Conlin, William Brooks, Blake Waring.

**Writing – original draft:** Matthew Conlin.

**Writing – review & editing:** Matthew Conlin, Gregory Dusek, John Ratcliff, John A. Callahan, Karen E. Kavanaugh, William Brooks, Blake Waring, Analise Keeney, William Sweet, Matthew J. Widlansky.

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
