## [Decision Letter · Decision Letter 0]

28 Nov 2025

PONE-D-25-29033Filling the gaps between tide gauges: Demonstrating high-resolution seasonal high tide flooding predictions using NOAA’s Coastal Ocean ReanalysisPLOS ONE

Dear Dr. Conlin,

Thank you for submitting your manuscript to PLOS ONE. After careful consideration, we feel that it has merit but does not fully meet PLOS ONE’s publication criteria as it currently stands. Therefore, we invite you to submit a revised version of the manuscript that addresses the points raised during the review process.

We look forward to receiving your revised manuscript.

Kind regards,

Fausto Cavallaro, PhD

Academic Editor

PLOS ONE

Journal Requirements:

“I have read the journal's policy and the authors of this manuscript have the following competing interests: Authors MC, JR, and JC are employed by Ocean Associates, Inc. Author BW is employed by Consolidated Safety Services, Inc.”

We note that one or more of the authors are employed by a commercial company: Consolidated Safety Services, Inc.

4. We note that Figure 1 and 7 in your submission contain map/satellite images which may be copyrighted. All PLOS content is published under the Creative Commons Attribution License (CC BY 4.0), which means that the manuscript, images, and Supporting Information files will be freely available online, and any third party is permitted to access, download, copy, distribute, and use these materials in any way, even commercially, with proper attribution. For these reasons, we cannot publish previously copyrighted maps or satellite images created using proprietary data, such as Google software (Google Maps, Street View, and Earth). For more information, see our copyright guidelines: http://journals.plos.org/plosone/s/licenses-and-copyright.

1. You may seek permission from the original copyright holder of Figure 1 and 7 to publish the content specifically under the CC BY 4.0 license.

5. We note that there is identifying data in the Supporting Information file “S2 Table.xlsx”. Due to the inclusion of these potentially identifying data, we have removed this file from your file inventory. Prior to sharing human research participant data, authors should consult with an ethics committee to ensure data are shared in accordance with participant consent and all applicable local laws.

-Location data

Additional Editor Comments:

The paper needs a minor revision.

Reviewers' comments:

Reviewer's Responses to Questions

**Comments to the Author**

1. Is the manuscript technically sound, and do the data support the conclusions?

Reviewer #1: Yes

Reviewer #2: Yes

2. Has the statistical analysis been performed appropriately and rigorously?

Reviewer #1: Yes

Reviewer #2: Yes

3. Have the authors made all data underlying the findings in their manuscript fully available?

Reviewer #1: Yes

Reviewer #2: Yes

4. Is the manuscript presented in an intelligible fashion and written in standard English?

Reviewer #1: Yes

Reviewer #2: Yes

5. Review Comments to the Author

Reviewer #1: The article is well written and tackles the important topic of predicting risks of coastal flooding for months ahead. The frequency of these events is rapidly increasing along all coastal locations, and particularly for locations with high rates of relative sea level rise. Several of the authors had worked on and published a related paper presenting a new method to predict monthly high tide flooding. The previous work was somewhat limited spatially to tide gauge locations. The present work extends substantially the spatial span of the predictive model by using the relatively recent oceanographic CORA reanalysis data set.

The new method extends the previous work of estimating daily cumulative flood probabilities for tide gauges based by using the CORA data. The article makes a well documented argument that the CORA reanalysis data set can be used to predict monthly high tide flooding, at least for most locations.

I have two reservations regarding the paper. The authors are very knowledgeable and already discuss these limitations on a few occasions but, if the authors agree, I think these limitations should be pointed out more explicitly including in the discussion and conclusion. These limitations were already present when the method only focused on tide gauge data but the new method based on CORA reanalysis increases greatly the spatial extent of the predictions and now includes more locations with complex bathymetry within shallow bays, lagunas, and estuaries. Figure 7 provides a good example of how the CORA based method expands substantially the area for the predictions.

(1) The spatial resolution of the CORA data set is 400-500m. This resolution is not a concern for the open coast and large deep bays but I think this resolution is too coarse to accurately model some locations in bays and estuaries, particularly where water level dynamic is influenced by shallow passes with widths in the range of a few meters to even 10-20m. If this is indeed a limitation, the article should include a more specific discussion that the method should be used carefully and possibly not at all for shallow coastal areas including those influenced by shallow passes. Some related precautions were taken for the modeling example near Charleston using points 500m away from the shoreline and I agree with the related great benefits discussed on lines 490-493. If the authors agree I think they need to be more explicit maybe stating in the conclusion that the method should be used only for coastal areas where modeling at a 400-500 m resolution does not significantly influence water level dynamic. It could be added that a reanalysis at a higher resolution would narrow this limitation.

(2) The method does not work as well in the Gulf, Eastern (EG) and Western Gulf (WG). This is not a big concern given the benefits, but I would like to have more details on the locations for which the assessment was made and a more extensive and more precise discussion as to areas of the Gulf where the method should not be considered as accurate and possibly not be applied. The limitations are clearly shared and discussed for the tide gauges of Grande Isle, Louisiana and Rockport, Texas in Appendix S4. I would like to see more explicit results for all the locations of the EG and WG similar to the Table in appendix S1. In particular, I am wondering how many locations were used to compute the EG and WG averages. For the WG, the tide gauges of Corpus Christi and Sabine Pass were removed for data quality concerns (line 212) as well as Rockport (line 381) "where errors in CORA water levels and tide predictions led to strong overprediction of HTF probabilities". I would like to know more explicitly which stations were left for the assessment, clearer information than presently in the text and in Appendix S2.

Other less important to me detail type question regarding lines 102-110. The method was setup for tide gauges, i.e. for locations that were selected to typically not go dry and benefit from some water depth during most of the year. The CORA data set allows to expand the predictions to coastal areas including some which will, at times, have very little water depth. For shallow areas, the tidal dynamic and potential nonlinear interactions between the non tidal component and the tidal dynamic could affect the distributions, e.g. the timing of the tides will change based on non tidal water level component. Also could this affect the estimation of the dampening for locations with at times very low water levels? As the work focusses on the higher deciles of water levels this is likely not significant but wondering how relevant it is and if it should be further discussed. Note that the largest differences between CORA and tide gauged based predictions are for low water level predictions and I am concerned about areas that have, at times, water levels lower than that of nearby tide gauges and may not be as accurately modeled using CORA.

In line 284-285, the authors share that "The RMSE of CORA-derived trends relative to the published values at the NWLON stations was 1.27 mm/yr, however not considering Eagle Point, TX the RMSE was 0.74 mm/yr (S3 Appendix)." Eagle Point should not be used for rslr trends during that period as it was strongly impacted by oil and gas extraction leading to large changes, increasing and decreasing, in land subsidence and a definitely non linear rslr trend (there is a paper documenting rslr changes and their driver for that location, I can find it if helpful).

Line 587: Great to mention future modeling work to provide calibrated uncertainty bounds on HTF predictions. Big endeavor but would be most helpful to the users.

Reviewer #2: This manuscript assesses the ability of a water level reanalysis, CORA, to reproduce high tide flooding (HTF) predictions generated from tide gauge (TG) data (using the methodology of Dusek et al 2022). A pilot example of the degree of spatial variability of high tide flooding metrics around Charleston, South Carolina is presented. Finally, the authors give an example of how uncertainty in CORA might be introduced into the HTF predictions.

This is a "methods" paper, and is a useful analysis that may help serve to extend the spatial coverage of HTF predictions. The analysis is original, well-described, and clearly executed -- so is technically sufficient for publication. I have 4 substantive comments that would improve the paper. I am not wed to a particular approach to achieve these goals, but I think they should be carefully considered by the authors.

Substantive issue 1. The danger in this analysis is that CORA does well at tide gauge locations (which are constrained by data assimilation), but poorly elsewhere, where data assimilation is a weaker constraint. The authors do a good job showing that there are not dramatic differences in CORA v. TG HTF predictions at assimilating and non-assimilating tide gauges, which helps their argument. However, there are more assimilating tide gauges than non-assimilating, and substantial differences in the spatial distribution of non-assimilated tide gauges evident in Fig. 1. For example there seem to be a lot of non-assimilated tide gauges in the Chesapeake Bay, and very few over the entire SAB. In addition, the tide gauges that aren't assimilated appear to be upstream and in problematic locations. In some sense this might be a *good* thing, because CORA seems to work even in these problematic sites. EXCEPT for Grand Isle and Rockport, which are thrown out of the analysis. A physical understanding of these differences across non-assimilating TGs would help inform where the use of CORA might be problematic. I have the following suggestions that the authors might pursue to bring these concerns to light, and, if possible, alleviate them:

a. Provide more details about the non-assimilated TG comparison. Explain the rationale for which tide gauges are assimilated and why, and whether we think the non-assimilated TG's are representative. Perhaps include another figure which shows more detailed results at non-assimilating sites (also see my thoughts on substantive issue 2).

b. Carefully choose words about what is demonstrated by the non-assimilated gauges (I think for the most part the authors did a good job).

c. There should be some mention in the discussion about the potential limitations of CORA, and potential next steps that could truly support confidence in predictions at non-TG locations. A true out-of-sample validation exercise would be a difficult task as it might need a non-data assimilating appropriate hydrodynamic model, or by some other on-the-ground metric of flooding. I don't think this is necessary in order to publish this paper but some discussion of limitations and possible alternative validation exercises would help. (I wonder if there is a free running version of CORA that could be used to assess the effect of assimilation on water level variability away from TGs?).

d. The appendix discussing Grand Isle and Rockport analyses how the statistics differ but does not really address why. A more complete description of the processes underlying differences, including their spatial scale, would help the reader (or future implementers) decide what generic types of locations might be problematic.

Substantive issue 2. My second comment concerns the lack of clarity what driver of sea level variability (linear SLR, tides, etc) is causing the discrepancy between CORA and TGs in different locations. I am very interested in seeing these deviations by tide gauge location (similar to what is done in Appendix 4). It looks like appendix S3 is a good start. At the very least, I would consider moving that figure into the main text. You could also add more detail, i.e. color coding points by region, and using square and circles (as in figure 1) to differentiate between assimilating and non-assimilating tide gauges. Note that even if the RMSEs/biases are calculated it's difficult to identify how these cascade into errors in flood probabilities.

Substantive issue 3. How does vertical land motion at tide gauges enter into CORA? Is it all driven by the data assimilation at tide gauges, and thus contains both an ocean steric/mass change and VLM? I think it is worth including some text into this paper on where those changes arise. They have very different implications for the spatial scales over which long term rates (or even nonlinear) changes can be extrapolated. It would also help interpret the deviations shown in Fig. S3.

Substantive issue 4. I would prefer to have the entire analysis/results communicated using the probabilistic approach rather than just an add-on at the end (in fact, a probabilistic propagation of uncertainty should probably be used for the TG-based HTF predictions as well). However, if the paper stays in the same format, the methodology for the probabilistic presentation of results (Figure 9) should be conveyed in the methods section, even if just sketched out. There needs to be at least a few equations describing which terms in the HTF model are assigned uncertainties, and how those are assigned.

Other minor comments are below:

30: the basis for -> an approach to generating

36: remove also

84: see line 30 comment

112: "SLR-adjusted" I assume this means removing the linear trend, but it's not defined explicitly and could be confusing.

132-134: I think this paper is using the p(t) term, but this sentence confused me. Can you be explicit that it is included?

146-153: worth including some details about the offshore boundary conditions, forcing, and source of linear trends here.

208-209: Are these sometimes dry locations used in the analysis, and which are they?

213-214: note smaller fraction of unassimilated than assimilated. And note the spatial distribution.

278-279: this sentence confused me. perhaps could be dropped?

293: "generalize these likelihood categories" -- could you remove?

371-382: units for bias and MAE

465-493: I think the inclusion of time series at two representative locations, for each application, would help the reader understand what was happening to modify the HLs.

493: is "morphodynamics" the right word? hydro- instead?

511 and 526: can you use a better word than "strong"?

511-534: here is where substantive issues 1 and 2 might be discussed in more detail, if not earlier in the results section.

590-593: see substantive issue #4.

6. PLOS authors have the option to publish the peer review history of their article (what does this mean?). If published, this will include your full peer review and any attached files.

Reviewer #1: No

Reviewer #2: No

---

## [Author Response · Author response to Decision Letter 1]

12 Jan 2026

Please see the attached Cover Letter and Response to Reviewers files.

---

## [Decision Letter · Decision Letter 1]

25 Feb 2026

Filling the gaps between tide gauges: Demonstrating high-resolution seasonal high tide flooding predictions using NOAA’s Coastal Ocean Reanalysis

PONE-D-25-29033R1

Dear Dr. Conlin,

We’re pleased to inform you that your manuscript has been judged scientifically suitable for publication and will be formally accepted for publication once it meets all outstanding technical requirements.

Kind regards,

Fausto Cavallaro, PhD

Academic Editor

PLOS One

Additional Editor Comments (optional):

The authors addressed all the reviewers comments. The paper now results improved and it can be accepted

Reviewers' comments:

Reviewer's Responses to Questions

**Comments to the Author**

1. If the authors have adequately addressed your comments raised in a previous round of review and you feel that this manuscript is now acceptable for publication, you may indicate that here to bypass the “Comments to the Author” section, enter your conflict of interest statement in the “Confidential to Editor” section, and submit your "Accept" recommendation.

Reviewer #1: All comments have been addressed

Reviewer #2: All comments have been addressed

2. Is the manuscript technically sound, and do the data support the conclusions?

Reviewer #1: Yes

Reviewer #2: Yes

3. Has the statistical analysis been performed appropriately and rigorously?

Reviewer #1: Yes

Reviewer #2: Yes

4. Have the authors made all data underlying the findings in their manuscript fully available?

Reviewer #1: Yes

Reviewer #2: Yes

5. Is the manuscript presented in an intelligible fashion and written in standard English?

Reviewer #1: Yes

Reviewer #2: Yes

6. Review Comments to the Author

Reviewer #1: Thank you for the response to our comments and the modifications to the paper including having more extensive discussions of the limitation of the approach/CORA for some parts of the coast including shallow areas. I appreciate the updated tables and including N. Having Figure 2 early in the paper was also an excellent idea. The paper works well for me now. I like the idea of analyzing pre- and post-assimilated CORA output "to deduce a more robust understanding of the spatially variable effects of the data assimilation as a function of morphodynamic environment and/or distance from NWLON stations". I hope you get a chance to work on that question. Keep going with this project, this is highly valuable for coastal communities (provided that the limitations are well understood) and will probably be extensively used.

Reviewer #2: I am satisfied with the authors' responses to my review.I am satisfied with the authors' responses to my review.

7. PLOS authors have the option to publish the peer review history of their article (what does this mean?). If published, this will include your full peer review and any attached files.

Reviewer #1: No

Reviewer #2: No

---

## [Editor Report · Acceptance letter]

PONE-D-25-29033R1

PLOS One

Dear Dr. Conlin,

I'm pleased to inform you that your manuscript has been deemed suitable for publication in PLOS One. Congratulations! Your manuscript is now being handed over to our production team.

Kind regards,

on behalf of

Professor Fausto Cavallaro

Academic Editor

PLOS One